# Psychologists’ Role in Concussion Assessments for Children and Adolescents in Pediatric Practice

**DOI:** 10.3390/ijerph17207549

**Published:** 2020-10-17

**Authors:** Roger W. Apple, Brian M. Stran, Brandon Tross

**Affiliations:** Homer Stryker M.D., School of Medicine, Western Michigan University, Kalamazoo, MI 49008, USA; brian.m.stran@wmich.edu (B.M.S.); brandon.e.tross@wmich.edu (B.T.)

**Keywords:** concussion assessment, concussion assessment in pediatric practice, role of psychologist in concussion assessment

## Abstract

An estimated 1.1 to 1.9 million children and adolescents in the United States are treated for a sports- or recreationally-related concussion each year. The importance of formalized assessment and measurement of concussion symptoms has been widely recognized as a component of best-practice treatment. The present paper reviews a sample of the most commonly used measures of concussion symptomology and explores psychologists’ role in their application in a pediatric practice. In addition, other issues such as accessibility and the appropriateness of application with child and adolescent patients are discussed. Literature is reviewed from journals pertaining to pediatric and adolescent medicine, sports medicine, neuropsychology, and testing and measurement.

## 1. Introduction

Psychologists’ role in pediatric practice has been steadily growing, particularly with the development of integrated behavioral health (IBH) services [1]. Having psychologists integrated into pediatric practice makes them invaluable members of the medical team, providing psychological services that previously would have required a referral to an outside source. These outside sources include community mental health centers, private practice psychologists, or private neuropsychologists. Although these outside sources may be able to provide excellent services in a traditional mental health setting, many patients will not follow through on the referral. The reasons patients often do not follow through on these outside referrals are numerous but include factors such as the stigma associated with needing a mental health service, not wanting to go to another office, and confusion around navigating mental health services. The advent of IBH has allowed psychologists to work as collaborative members of the medical team providing critical information and data, such as concussion assessment data, in real time. Working as integrated members of the medical team often helps reduce or even eliminate the stigma associated with psychologists and mental health services as well as eliminating the need to travel to another location.

WMed Health, the clinical services side of Western Michigan University, Homer Stryker M.D. School of Medicine in Kalamazoo, Michigan, USA, has developed an IBH program in its department of pediatric and adolescent medicine. The WMed IBH model of integrating psychologists into pediatric practice has many benefits: it allows the physician access to reliable data gathered from psychologists trained in the administration, scoring, and interpretation of assessments; allows for more in-depth assessments; increases clinic efficiency by allowing the physicians to continue to see additional patients while the psychologists are collecting assessment data; creates a single point of access for multiple services; and ultimately leads to improved patient outcomes. This review discusses the role of psychologists and neuropsychologists in concussion assessments for children and adolescents in pediatric practice, as well as identifying and describing some of the most commonly used concussion assessments.

At WMed Health, the IBH team consists of one licensed psychologist who supervises licensed master’s level psychology interns who are pursuing their Ph.D. at Western Michigan University. Being located in a large university allows us to provide IBH services to most of our primary care and some subspecialty clinics. However, IBH team members work collaboratively with our physicians, nurses, medical assistants, medical residents, and medical students. Other sites providing IBH services may also use social workers and licensed professional counselors.

## 2. Role of Psychologists

As psychologists continue to become more integrated within primary care settings, their contributions to patient care are expanding, resulting in improved care for a diverse range of clinical presentations. Psychologists integrated in primary care settings can often take on several different roles, including behavioral health consultants, psychotherapists, integrated behavioral health consultants, and test administrators [2]. Although there is little research on the role of psychologists in pediatric concussion care, many physicians recognize the contributions of integrated psychologists as needed to improve the overall care of patients [3]. Outside concussion treatment, psychologists in primary care settings play significant roles in the prevention, identification, and management of illnesses like heart disease and cancer [4]. Additionally, psychologists in primary care settings are in the position to collaborate with physicians in the treatment of co-occurring mental health disorders that may exacerbate physical health problems, like diabetes and obesity [5]. Because the diagnosis and treatment of concussions involve similar interdisciplinary concerns, psychologists within primary care settings have the potential to aid physicians in the care of pediatric concussion patients.

Whereas primary care providers are often responsible for the diagnosis and treatment of concussions, psychologists can also play an important role in providing comprehensive care for patients. With regard to the identification and management of concussions and concussive symptoms, psychologists’ training and ability to administer and interpret cognitive assessments can be crucial assets in delivering a high standard of care to patients. Psychologists do not always need to administer all assessments; a multitude of clinicians, athletic trainers, occupational therapists, physicians, and nurse practitioners can also administer many of these assessments. In addition to assessment, psychologists may also contribute to the treatment and management of concussions by providing interventions to increase treatment compliance and prevent future head injury. The present section explores the potential benefits that psychologists may have in the identification, diagnosis, treatment, and management of concussion and concussive symptoms. 

Psychologists’ time with patients can be highly variable, ranging from a few minutes, to administer an assessment and share the results with a physician, to several hours for a comprehensive psychological or neuropsychological evaluation. The psychologist-to-patient ratio to achieve benefit for the pediatric practice is also highly variable due, in large part, to the high demand, the sparse availability of pediatric psychologists and neuropsychologists, and the willingness of pediatric practices to hire optimal staffing for integrated behavioral health services. Many pediatric practices might share one psychologist, extremely limiting access to that person, whereas other practices have a psychologist or behavioral health consultant see every patient on the schedule, just like a nurse or medical assistant.

### 2.1. Identification and Diagnosis

In the WMed Health model, patients are identified to be seen by a psychologist in three ways: (1) internal or external referral from a physician directly to the pediatric psychology program, (2) review of the pediatric clinic schedule by IBH staff during morning and afternoon huddles to identify potential patients with mental health needs, and (3) the physician identifying a mental health need during an appointment and requesting the psychologist to see the patient in real time.

Although the administration of complex neurocognitive assessment batteries should be left to specialized neuropsychologists, general practicing psychologists have the ability to provide a wide range of assessments and measures to improve diagnosis and track progress [6]. Although many concussion screening tools can be used by most medical professionals [7,8,9,10,11,12,13,14,15,16,17,18,19,20,21,22,23,24,25,26,27,28], psychologists have unique training in cognitive assessments that may aid them in achieving more accurate and consistent results [29]. Contributing to the accuracy of the results, psychologists are trained in achieving uniform administration, as well as in assessing psychosocial factors that may provide false positives or false negatives in results, like lack of sleep and malingering [29]. In addition to screening tools, psychologists have the unique ability to conduct more complex assessments, including measures of intelligence and academic ability that may be instrumental to understanding the severity and impact of a concussion [30]. As difficulties like intellectual disabilities and academic problems may impact the presentation of concussion symptomology [31], psychologists can aid in the identification of differential diagnoses and exacerbating factors. Beyond the administration of assessments and measures, many psychologists are trained in the identification of measures and tests that fit the needs of the situation [32]. Given that each measure and assessment has strengths and limitations, the results of a measure may be inaccurate or invalid if inappropriately administered or given to a patient outside the measure’s scope [33]. Through this process, psychologists can aid primary care settings in identifying, administering, and interpreting the most appropriate measures to be used for each patient. Table 1 [7] provides an adapted list of commonly used scales and assessment tools used in the identification of concussion symptomology. The training required to administer these assessments varies greatly. 

### 2.2. Treatment and Management

In addition to assessments, psychologists working within primary care setting can assist physicians in the treatment and management of concussions and associated symptoms. One way psychologists have been identified as helpful in primary care settings is in improving treatment compliance through psychological interventions [34]. Psychologists may aid patients and families in making plans to follow physicians’ recommendations, like minimizing screen time and avoiding excessive physical activity. This process may also involve troubleshooting potential barriers to treatment compliance and managing environmental stressors, as well as building trust and motivation to comply with physicians’ recommendations. Another way in which psychologists in primary care settings can aid patients with concussions is helping patients and caregivers prevent future concussions [34]. This might include identifying dangerous situations or sports positions and developing strategies to minimize risk. Lastly, psychologists in primary care settings are often equipped with the skills to provide treatment for the psychological symptoms associated with concussions, like anxiety and depressionogenic thinking patterns [35]. The treatment of these psychological symptoms is particularly important in the management and treatment of the patient’s symptoms due to the association of anxiety and depressive symptoms with an increased risk of prolonged concussive symptoms [36,37]. These psychological interventions are particularly important for pediatric care due to the salience of family systems and environmental concerns when working with children and adolescent patients. Caregivers benefit from having behavioral strategies for managing children’s noncompliance in treatment, a plan for changing environmental risks, and support for addressing comorbid mood and anxiety symptoms. Although the role of providing reliable and valid assessments of concussion symptoms is important, psychologists’ ability to assist physicians in the treatment and management of concussion symptoms is an equally crucial service that primary care providers can offer to patients and their families. 

## 3. Role of Neuropsychologists

Trained in understanding the psychological underpinnings of emotion, behavior, neurodevelopment, and brain injury from a neurological perspective, neuropsychologists are positioned to serve a unique role in integrated behavioral health teams in the assessment and management of pediatric concussions [38,39,40,41]. Although there is a paucity of research focused on pediatric patients with concussions under the age of 14, neuropsychological research has focused on understanding the role of assessment in the management and treatment of pediatric concussions [42]. This section will provide an overview of the role of neuropsychologists in the pediatric integrated behavioral health team. 

Pediatricians are often tasked with managing the care of patients who have suffered a concussion; however, in a recent survey, only 45% of health care providers felt “very prepared” about making decisions regarding when to return to learning or return to sports [43]. The Concussion in Sports Group [44] highlighted the importance of using objective neuropsychological assessment results in making return to school and return to play decisions. When making return to play and return to sport decisions, 24.6% of healthcare providers reported “seldom” using objective screening or assessment and 22% reported “never” using objective measures [43]. Furthermore, there is wide documentation of healthcare providers providing inadequate education to pediatric patients and their families related to managing concussive events [45,46]. Although some healthcare providers may not be aware of the tools available to use in concussion assessments and feel unprepared to competently manage the care of pediatric patients with mild traumatic brain injuries (mTBIs), neuropsychologists on the integrated behavior health teams can readily fill these gaps.

Having neuropsychologists working on multidisciplinary teams is not novel; however, recently, there has been a burgeoning body of research exploring the role of neuropsychologists on integrated health teams [47,48,49]. In a recent study completed by Kubu et al. [48], 15.3% of surveyed neuropsychologists indicated they worked in a primary care setting that included pediatrics, primary care, internal medicine, and geriatric medicine. Neuropsychologists working on integrated health teams highlighted five areas where they made the most notable contributions to the integrated health team: (1) recommendations/prognosis (i.e., return to learning, return to play), (2) communication with the team (i.e., interpretation of neuropsychological reports, placing the patient in a broader context using psychosocial variables), (3) diagnosis, (4) medico-legal, and (5) communication with patients and their families [48]. The aforementioned areas of impact are important to the efficacious care and management of mTBI. 

A neuropsychological evaluation consists of integrating information from numerous sources obtained through clinical interviews and reviews of the patient’s medical records, as well as the administration of standardized assessments that evaluate various aspects of brain functioning [50]. Neuropsychological tests can be administered by traditional paper and pencil methods or via computerized measures [51]. When pediatric patients present to the clinic after suffering an mTBI, a neuropsychologist may elect to conduct a cognitive screening assessment, a brief neuropsychological assessment, or a comprehensive neuropsychological assessment. 

### 3.1. Cognitive Screening 

Cognitive screening is used to form diagnostic accuracy and identify the presence of medical emergencies [40]. The Standardized Assessment of Concussion (SAC) is a brief screening tool administered by athletic trainers that is often used to assess altered mental status 48–72 h post-injury. Following the cognitive screening, neuropsychologists, along with collaboration with a physician or nurse practitioner, can also provide developmentally appropriate education to patients and their families that includes (1) assurance that most children who experience an mTBI recover within 60–90 days [52], (2) common symptoms children experience after a concussion, (3) signs to look for that may indicate a more severe injury, and (4) an outline of the return to school and return to play process. The current literature indicates that physicians have often advised return to play and school too soon and have provided patients and families with insufficient education in managing their symptoms [53]. Neuropsychologists can help reduce adverse consequences due to lack of education and poor decision-making regarding resuming a normal level of activity too soon.

### 3.2. Brief Neuropsychological Assessment

The purpose of the brief neuropsychological assessment (BNA) is to gain a better understanding of the patient’s symptoms [40]. Although most children’s symptoms will resolve within a week, some children report lingering symptoms for up to a year after injury [54,55]. Post-concussive symptoms are said to be non-injury-related and have been linked to premorbid learning; affective, somatic, and behavioral problems; feigning; parental anxiety; and family stress. If the patient continues to experience post-concussive symptoms more than two weeks after their report injury, a brief neuropsychological assessment (BNA) may be administered. At this phase, the BNA would provide more information than a screening tool but less information than a comprehensive neuropsychological assessment [40]. As part of the BNA, a thorough clinical interview is critical to generate hypotheses about what may be maintaining the child’s symptoms, as an mTBI may exacerbate preexisting conditions causing a prolonged recovery [56,57,58]. Due to their sensitivity shortly after an mTBI, the BNA may include tests measure executive functioning, memory and attention, and psychomotor and processing speed, as well as instruments that are insensitive to mTBI [40,41,42,43,44,45,46,47,48,49,50,51,52,53,54,55,56,57,58,59].

### 3.3. Comprehensive Neuropsychological Assessment

Although most children become asymptomatic within four weeks, at three months post-mTBI, between 13% and 29% of children continue to experience post-concussive symptoms and 2.3% of children remain symptomatic after a year [44,56,60,61]. When the patient continues to demonstrate post-concussive symptoms four months after their injury, they are in the long-term recovery phase [40]. The purpose of assessment at this point in recovery is to explore and identify the causes for symptom maintenance, diagnostic accuracy, and the development and implementation of an individualized treatment plan [40]. Several factors affect the prolongation of mTBI-related symptoms such as prior concussions; premorbid attentional issues and learning disabilities; past medical history of poor sleep hygiene, headaches, anxiety, depression, and familial stress; and litigation [62,63]. Patients often attribute their post-concussive symptoms to mTBI when they were present before their injury [56].

### 3.4. Performance Validity Testing

Performance validity tests (PVTs) are standalone tasks that are embedded within the neuropsychological assessment, which may appear difficult but are easy to complete [64]. With between 15% and 20% of pediatric patients putting forth insufficient effort on neuropsychological assessments, PVTs provide neuropsychologists with objective evidence that their patient’s performance was either a valid or invalid measure of their abilities [65,66]. Performance on PVTs has been shown to be unrelated to neurological impairment and intellectual status [67]. Use of PVTs are beneficial because they assist in ruling out neurological injury and focus attention on identifying the non-neurological causes of the patient’s symptoms [68].

The usefulness of neuropsychological assessments in the management of concussions has been highlighted since the first Concussion in Sports Group (CISG) meeting and continues to be recognized as a salient component in patient care [44,69]. Using neuropsychological tests, neuropsychologists can help identify neurological injury or bring attention to non-neurological factors elongating post-concussion symptoms. Beyond assessment, the role of the neuropsychologist also includes providing developmentally appropriate psychoeducation to the patient and the family, consulting with medical professionals to make to return to school and return to play decisions, interfacing with educators and coaches to ensure a smooth integration back to school and sports, and assisting in identifying beneficial interventions to mitigate and eliminate post-concussive symptoms. 

There is added benefit to having a neuropsychologist on the behavioral health team. Patients are most likely to follow through with a neuropsychological evaluation when the neuropsychologist is a part of the treatment team. When physicians refer their patients to a neuropsychologist within an integrated behavioral health clinic using a warm handoff, the patient is more likely to follow through with the appointment than if referred to a neuropsychologist in another department [47]. 

## 4. Conclusions

Psychologists integrated into pediatric practice are able to offer their expertise in concussion assessments. This is a tremendous benefit to pediatric practice because psychologists can not only conduct concussion assessments, they can also provide expanded assessment capabilities that the medical staff would not have time for, as well as provide assistance with prevention, identification, and management of multiple illnesses beyond concussions such as cancer, obesity, heart disease, and diabetes. Psychologists can also help to identify barriers and confounding psychosocial variables to treatment. Notably, the availability of pediatric psychologists and neuropsychologists is limited in academic institutions and even more in non-academic centers. Each organization will need to determine how best to use pediatric psychologists and neuropsychologists, with some clinics requiring referrals, use in subspecialties only, or only in persistent concussion clinics. University settings, like WMed Health, have the luxury and availability of psychology trainees to help fill these roles.

Neuropsychologists integrated into pediatric practice offer an even more specialized role focusing on brain injury and neurodevelopment. Neuropsychologists often integrate large amounts of data, including concussion assessment data, as well as conduct more comprehensive neuropsychological evaluations when needed and collaborate with physicians to determine the most appropriate return to learning and/or sports. Both psychologists and neuropsychologists play a significant role in helping patients with non-neurological causes of symptoms.

The implications of having psychologists and neuropsychologists integrated into pediatric practice reach far beyond concussion assessments. Integrating a multidisciplinary team of providers into pediatric practice provides numerous benefits such as single-point-of-care services, real-time intervention from multiple team members, greatly improved communication among providers, and ultimately improved patient outcomes.

## Figures and Tables

**Table 1 ijerph-17-07549-t001:** Concussion assessments.

Assessment	Type	Age Range	Description
Glasgow Coma Scale (GCS)	Assessment tool	>2 years	The GCS is a measure of impairment of consciousness level resulting from a head injury. The scale measures eye response, verbal response, and motor response as measures of impairment.
Pediatric Glasgow Coma Scale (PGCS)	Assessment tool	≤2 years	PGCS is a measure of impairment of consciousness level resulting from a head injury in children. The scale measures eye response, verbal response, and motor response as measures of impairment.
Standardized Assessment of Concussion (SAC)	Assessment tool	≥9 years	SAC is a short measure of concussion impairment. The scale includes measures of orientation, memory, and concentration.
Sport Concussion Assessment Tool 3 (SCAT-3)	Assessment tool	≥13 years	SCAT-3 is the third iteration of the tool developed by the International Symposium on Concussion in Sport. The tool includes evaluations of symptoms, cognitive impairment, and physical impairment
Sport Concussion Assessment Tool 5 (SCAT-5)	Assessment tool	≥13 years	SCAT-5 is the fourth iteration of the tool developed by the International Symposium on Concussion in Sport. The tool is a measure of concussion symptoms.
Child Sport Concussion Assessment Tool 5 (CHILD SCAT-5)	Assessment tool	≤12 years	Child SCAT-5 is the second iteration of the tool developed by the International Symposium on Concussion in Sport made for children. Similar to the adolescent/adult version, questions were specifically designed for children.
Military Acute Concussion Evaluation 2 (MACE-2)	Assessment tool	≥18 years	MACE-2 is an acute assessment tool designed to assess concussion symptoms in service members. The assessment includes measures of concussion symptoms, cognitive ability, and neurological symptoms.
King–Devick Test	Assessment tool	≥5 years	The King–Devick test is a 2 min assessment of eye movement, attention, and language functioning. The assessment can be administered via tablet.
Balance Error Scoring System (BESS)	Balance test	≥9 years	BESS is a measure of static postural stability in patients with head injuries. Requires patient to stand in different positions.
Sensory Organization Test (SOT)	Balance test	≥18 years	SOT is a measure of a patient’s visual, proprioceptive, and vestibular symptoms resulting from a head injury. The assessment involves the patient standing up in six different conditions.
Acute Concussion Evaluation (ACE)	Symptom scale	≥5 years	ACE is a scale that includes a symptom checklist and identification of potential risk factors for complicated diagnosis.
Concussion Symptom Inventory (CSI)	Symptom scale	≥4 years	CSI is a measure of subjective concussion symptoms. Designed to monitor subjective concussion symptoms over time.
Graded Symptom Checklist and Graded Symptom Scale (GSCGSS)	Symptom scale	≥13 years	GSCGSS is a screening measure of concussion symptoms and severity over a 72 h post-injury span.
Health and Behavior Inventory (HBI)	Symptom scale	8 to 15 years	HBI is a self- and parent-reported measure of concussion symptoms.
Post-Concussion Symptom Inventory (PCSI)	Symptom scale	5 to 18 years	PCSI is a self-report for age-specific items regarding the cognitive, emotional, sleep, and physical domains of concussion symptomology.
Post-Concussion Symptom Scale (PCSS)	Symptom scale	≤11 years	PCSS is a screening measure of concussion symptoms, designed to track changes in symptoms over time.
Rivermead Post-Concussion Symptoms Questionnaire (RPCS)	Symptom scale	13 to 65 years	RPCS is a self-reported scale of 16 concussion symptoms across three domains: physical, cognitive, and behavioral.
Automated Neuropsychology Assessment Metrics (ANAM)	Computerized neurocognitive	≥16 years	ANAM is a computer-based battery of tests designed to measure neurocognitive skills, including attention, processing speed, and visuospatial ability.
Pediatric Automated Neuropsychological Assessment (PANAM)	Computerized neurocognitive	≥10 years	PANAM is a computer-based battery of tests designed to measure neurocognitive skills, including attention, processing speed, and visuospatial ability in patients younger than 10 years old.
CogSport/Axon	Computerized neurocognitive	≥11 years	Cogsport is a computerized neuropsychological test battery designed to measure psychomotor function, processing speed, vigilance, and memory.
Concussion Resolution Index (CRI)	Computerized neurocognitive	≥13 years	CRI is a computerized neuropsychological testing battery designed to measure processing speed, visual scanning, and memory.
Immediate Post-Concussion Assessment and Cognitive Testing (IPCACT)	Computerized neurocognitive	≥5 years	IPCACT is an online computerized neuropsychological test battery that includes the collection of demographic information, a list of symptoms, and measures of attention, memory, working memory, reaction time, and learning.

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
