# Peer review of "Psychologists’ Role in Concussion Assessments for Children and Adolescents in Pediatric Practice"

_ijerph, 2020, doi:10.3390/ijerph17207549_

Round 1

Reviewer 1 Report

This is a descriptive paper about the possible role of the psychologist or neuropsychologist on the multidisciplinary team involved with assessing and treating children and adolescents with concussion and the resultant complications.

This beneficial role cannot be disputed.

In the enthusiasm to convey this potential benefit the authors have not described any of the possible team members in such a clinic other than the physician.  It might be helpful to insert a one paragraph description of the full team of the Integrated Behavioural Health Program.

The role description in part 2 is quite accurate and the individual activities described are fine but to propose that psychologists and Neuropsychologists be the ones doing the Concussion Assessments is perhaps over enthusiastic.  Of the 23 tests listed the first 12 are tests designed to be administered at the sideline and in acute settings by a multitude of clinicians and athletic trainers.   The last 6 are Neurocognitive tests that can be optimally administered by psychologists but may also be used effectively and reliably by Occupational therapists.  I would also propose that the middle 5 are best done in the clinic by physicians or nurse practitioners.

I noted in 3.1 line 150 that the published title of the test is the Standardized Assessment of Concussion (McCrea 2001 - J.Athl.Train.) at tool administered ion their study by athletic trainers - not psychologists.

And I would see the item 3) signs to look for a more severe injury is a collaborative one with the physician or N.P..

Finally the availability of pediatric psychologists or neuropsychologists is so limited in our clinical environments we can only expect them to be available to work in our specialty subspecialty clinics or persistent concussion clinics and not in the community at large.

A good article but don't try to oversell the argument.

I would have liked to see some outcome data looking at improved care either in quality, function or efficiency.

Author Response

In the enthusiasm to convey this potential benefit the authors have not described any of the possible team members in such a clinic other than the physician.  It might be helpful to insert a one paragraph description of the full team of the Integrated Behavioural Health Program.

Paragraph describing the full team has been added.

The role description in part 2 is quite accurate and the individual activities described are fine but to propose that psychologists and Neuropsychologists be the ones doing the Concussion Assessments is perhaps over enthusiastic.  Of the 23 tests listed the first 12 are tests designed to be administered at the sideline and in acute settings by a multitude of clinicians and athletic trainers.   The last 6 are Neurocognitive tests that can be optimally administered by psychologists but may also be used effectively and reliably by Occupational therapists.  I would also propose that the middle 5 are best done in the clinic by physicians or nurse practitioners.

Information added that describes how various providers are able to administer many of these assessments.

I noted in 3.1 line 150 that the published title of the test is the Standardized Assessment of Concussion (McCrea 2001 - J.Athl.Train.) at tool administered ion their study by athletic trainers - not psychologists.

Error corrected

And I would see the item 3) signs to look for a more severe injury is a collaborative one with the physician or N.P..

Added information to make illustrate the collaboration.

Finally the availability of pediatric psychologists or neuropsychologists is so limited in our clinical environments we can only expect them to be available to work in our specialty subspecialty clinics or persistent concussion clinics and not in the community at large.

While we are very fortunate at WMed Health to have such access I agree that access is limited. I have added such information in the conclusion.

A good article but don't try to oversell the argument. I would have liked to see some outcome data looking at improved care either in quality, function or efficiency.

Yes, it was challenging to try to balance supporting psychologists and not overselling.

Reviewer 2 Report

In this review Dr. Apple and co-authors highlight an interesting progressive topic in the mental health care after concussion which have diversifying outcomes. I only have a minor remark as follows:

 In the introduction, it would be more informative if the authors had mentioned a general image of mental health services in order to define a specific context indicating later on the importance of the IBH and what are the points that Behavioral health services target other than a regular mental health service, since IBH is more integrative and informative when it comes to assessing outcomes of the concussion specifically by the psychologists.

Author Response

In this review Dr. Apple and co-authors highlight an interesting progressive topic in the mental health care after concussion which have diversifying outcomes. I only have a minor remark as follows:

 In the introduction, it would be more informative if the authors had mentioned a general image of mental health services in order to define a specific context indicating later on the importance of the IBH and what are the points that Behavioral health services target other than a regular mental health service, since IBH is more integrative and informative when it comes to assessing outcomes of the concussion specifically by the psychologists.

A significant addition was made to the first paragraph of the introduction describing more traditional mental health services which will help set the stage for the importance of IBH as suggested.

Reviewer 3 Report

The authors offer a review of the role of psychologists in pediatric practices to facilitate assessment of concussion and the consequences of concussion. The authors repeatedly state the value of psychologists in this regard throughout the manuscript, almost like an infomercial. However, they offer very little data. Also, the manuscript does no focus enough on concussion, but it discusses the other potential roles a psychologist might fill in a practice. Answers to the following questions would increase the value of the manuscript greatly:

  1. What is the appropriate psychologist to patient ratio to achieve benefit to the practice?
  2. How long should a psychologist spend with a patient?
  3. How are patients identified to be seen by a psychologist?
  4. How long does it take to perform each of the various psychological tests listed.
  5. Which are the best tests to use and when does one decide to use one test versus another?
  6. What outcome measures are used to validate the authors' opinion about the value of psychologist(s) in a practice? What is that data?
  7. What about school performance? Or return to physical education? Or return to play sport or other extracurricular activities? 

Author Response

The authors offer a review of the role of psychologists in pediatric practices to facilitate assessment of concussion and the consequences of concussion. The authors repeatedly state the value of psychologists in this regard throughout the manuscript, almost like an infomercial.

Thank you for the feedback. We were trying to support the need and importance of psychologists but it was challenging to balance support without overselling.

However, they offer very little data. Also, the manuscript does no focus enough on concussion, but it discusses the other potential roles a psychologist might fill in a practice. Answers to the following questions would increase the value of the manuscript greatly:

  1. What is the appropriate psychologist to patient ratio to achieve benefit to the practice?

This is a complicated question as some reviewers have already pointed out that pediatric psychologists are in very high demand which makes considering their use extremely important. I have added comments regarding this to section 2. Role of Psychologists. This was added in the third paragraph.

  1. How long should a psychologist spend with a patient?

A third paragraph was added to section 2. Role of Psychologists to discuss time with patient.

  1. How are patients identified to be seen by a psychologist?

A first paragraph has been added to section 2.1 regarding identification.

  1. How long does it take to perform each of the various psychological tests listed.

Many of the assessments used in pediatric practice have been designed to be completed within only a few minutes. Should a more comprehensive test be needed, a neuropsychological battery, or full neuropsychological evaluation and additional appointment with the psychologist or neuropsychologist would be necessary and could take several hours. However, the focus of this manuscript is working in the pediatric clinic as opposed to a co-located service. This additional information was added to section 2. Role of Psychologist

  1. Which are the best tests to use and when does one decide to use one test versus another?

We originally considered this very question but decided that it is beyond the scope of this manuscript, however, it is a question I often hear from medical providers. This supports having psychologists available who are trained in assessment in order to choose to correct tools to be used. Each case is different and may necessitate different tools. This is a great idea for future publications.

  1. What outcome measures are used to validate the authors' opinion about the value of psychologist(s) in a practice? What is that data?

Please review reference #3 which is an entire book regarding Behavioral Consultation in Primary Care and supports the need for psychologists.

  1. What about school performance? Or return to physical education? Or return to play sport or other extracurricular activities?

Return to school/sports/etc. was discussed in section 3 Role of Neuropsychologists

Round 2

Reviewer 3 Report

The authors have addressed many of the reviewers' comments. Details on test selection would have made the manuscript stronger, as would data on outcome.